# The Relationship between the Ewing Test, Sudoscan Cardiovascular Autonomic Neuropathy Score and Cardiovascular Risk Score Calculated with SCORE2-Diabetes

**DOI:** 10.3390/medicina60050828

**Published:** 2024-05-17

**Authors:** Andra-Elena Nica, Emilia Rusu, Carmen Dobjanschi, Florin Rusu, Claudia Sivu, Oana Andreea Parlițeanu, Gabriela Radulian

**Affiliations:** 1Diabetes Department, “Carol Davila” University of Medicine and Pharmacy, 050474 Bucharest, Romania; andra.nica@drd.umfcd.ro (A.-E.N.); carmen.dobjanschi@umfcd.ro (C.D.); claudia.topea@drd.umfcd.ro (C.S.); gabriela.radulian@umfcd.ro (G.R.); 2“Nicolae Malaxa” Clinica Hospital, 022441 Bucharest, Romania; 3“Doctor Carol Davila” Central Military University Emergency Hospital, 010825 Bucharest, Romania; florinrusumd@yahoo.com; 4“Marius Nasta” Institute of Pneumophysiology, 050159 Bucharest, Romania; oana_andreea@yahoo.com

**Keywords:** T2DM, Sudoscan, CAN, CVDs, SCORE2-Diabetes

## Abstract

*Background and Objectives*: Cardiac autonomic neuropathy (CAN) is a severe complication of diabetes mellitus (DM) strongly linked to a nearly five-fold higher risk of cardiovascular mortality. Patients with Type 2 Diabetes Mellitus (T2DM) are a significant cohort in which these assessments have particular relevance to the increased cardiovascular risk inherent in the condition. *Materials and Methods*: This study aimed to explore the subtle correlation between the Ewing test, Sudoscan-cardiovascular autonomic neuropathy score, and cardiovascular risk calculated using SCORE 2 Diabetes in individuals with T2DM. The methodology involved detailed assessments including Sudoscan tests to evaluate sudomotor function and various cardiovascular reflex tests (CART). The cohort consisted of 211 patients diagnosed with T2DM with overweight or obesity without established ASCVD, aged between 40 to 69 years. *Results*: The prevalence of CAN in our group was 67.2%. In the study group, according SCORE2-Diabetes, four patients (1.9%) were classified with moderate cardiovascular risk, thirty-five (16.6%) with high risk, and one hundred seventy-two (81.5%) with very high cardiovascular risk. *Conclusions*: On multiple linear regression, the SCORE2-Diabetes algorithm remained significantly associated with Sudoscan CAN-score and Sudoscan Nephro-score and Ewing test score. Testing for the diagnosis of CAN in very high-risk patients should be performed because approximately 70% of them associate CAN. Increased cardiovascular risk is associated with sudomotor damage and that Sudoscan is an effective and non-invasive measure of identifying such risk.

## 1. Introduction

Cardiac autonomic neuropathy (CAN) is a severe complication of diabetes mellitus (DM) strongly linked to a nearly five-fold higher risk of cardiovascular mortality [1]. Factors leading to diabetic complications, such as genetic predisposition, environmental signals, insulin resistance, immune dysfunction and inflammation have a similar impact on both microvascular and macrovascular complications [2]. Elucidation of the common mechanism of cell damage in DM put to rest any misconceptions that microvascular disease and macrovascular disease constitute distinct disease entities [3,4].

CAN represents a manageable complication of T2DM, posing an elevated risk of cardiovascular mortality [5]. Patients with CAN are often underdiagnosed due to limited physician awareness of the disease, its frequently late onset with mild or absent symptoms, and the absence of specific diagnostic tests for CAN [6]. Various cross-sectional studies have revealed a wide range of prevalence rates. For instance, studies have shown rates ranging from 16.7% in insulin-dependent diabetic patients participating in the Diabetes Control and Complications Trial (DCCT) cohort to as high as 60% in a community-based sample of type 2 diabetic patients over the age of 65 in Rochester, Minnesota [7,8].

Among the many diagnostic tools available, Ewing and Sudoscan stand out as promising methods for evaluating autonomic neuropathy and its implications for cardiovascular health. Annually, cardiovascular diseases (CVDs) account for approximately 17.9 million deaths, making up 31% of all global fatalities [9]. Public health policies must prioritize preventing cardiovascular issues and managing established risk factors like dyslipidemia, T2DM, and hypertension [10]. Cardiovascular diseases represent the primary contributors to both morbidity and mortality among individuals diagnosed with T2DM [11].

For patients over 40 with T2DM without coronary artery disease (ASCVD) or severe target organ damage (TOD), estimating the 10-year cardiovascular risk using the SCORE2-Diabetes algorithm is recommended. The 2021 ESC Guidelines also suggest using ADVANCE or DIAL models for such risk estimations in diabetic patients [12,13,14]. The current guidelines recommend using the SCORE2-Diabetes model and to estimate the individual 10-year risk of fatal and non-fatal cardiovascular events [15].

On the other hand, Sudoscan is a useful tool for detecting chronic microvascular complications of T2DM, which are additional cardiovascular risk factors. Its results can provide valuable information for diagnosing and monitoring neuropathy and assessing cardiovascular risk in diabetic patients [8].

As the diabetes epidemic progresses, the increasing prevalence of CAN poses significant life-threatening risks, including arrhythmias and silent myocardial ischemia, particularly impacting the cardiovascular health of patients with T2DM [16].

Managing T2DM demands a comprehensive strategy encompassing lifestyle adjustments and glycemic regulation, alongside mitigating cardiovascular risk factors [17]. Additionally, employing glucose-lowering medications known for their cardiovascular benefits, such as SGLT2 inhibitors [18] and GLP-1 receptor agonists [19], is essential.

This study aimed to explore the subtle correlation between the Ewing test, Sudoscan- cardiovascular autonomic neuropathy score, and cardiovascular risk calculated using SCORE 2 Diabetes in individuals with T2DM. By elucidating this correlation, we aim to shed light on new avenues for comprehensive cardiovascular risk assessment.

## 2. Materials and Methods

### 2.1. Study Desgin

This study was cross-sectional, effectuated between June 2019 and June 2020. Ethical approval for the study was obtained from the local Ethics Committee of “Nicolae Malaxa” Clinical Hospital. Informed consent was obtained from all participating patients.

### 2.2. Study Population

**Inclusion Criteria:** Individuals diagnosed with Type 2 Diabetes Mellitus (T2DM) and those who were overweight/obese aged between 40 and 69 years were included.

**Exclusion Criteria:** Patients who did not provide informed consent, had other types of diabetes (type 1 diabetes, latent autoimmune diabetes in adults, maturity-onset diabetes of the young), were outside the age range of 40 to 69 years, pregnant women, those diagnosed with neoplasms within the past five years, had stroke sequelae, a history of myocardial infarction, pelvic limb amputations, pre-existing chronic kidney disease before diabetes diagnosis, and neuropathy from alternative causes (e.g., alcoholism, vitamin B12 deficiency). Type 1 diabetes, latent autoimmune diabetes of adults, and maturity-onset diabetes of the young were diagnosed by performing specific autoantibody tests. The presence of these antibodies led to the exclusion of patients from the study.

**Variables:** In our article the primary outcome variable was the correlation between the Sudoscan-cardiovascular autonomic neuropathy (CAN) score and the cardiovascular risk calculated using the SCORE2-Diabetes algorithm. This study aimed to investigate the potential association between sudomotor dysfunction, as measured by the Sudoscan-CAN score, and increased cardiovascular risk in individuals with Type 2 Diabetes Mellitus (T2DM).

**Examination of patients:** Anthropometric indices recorded included height, weight, body mass index (BMI), waist circumference, and waist-to-hip ratio. Blood pressure values in supine and orthostatic positions, heart rate, and smoking status were also documented.

**Measurement of biochemical parameters:** The following samples were collected from venous plasma after 8 h of fasting: serum glucose, glycated hemoglobin HbA1c, total cholesterol, high-density lipoproteins cholesterol (HDLc), LDL cholesterol, triglycerides, bilirubin, C-reactive protein, serum creatinine, and electrolytes (potassium, magnesium, chloride, sodium, calcium) urea, liver enzymes: aspartate aminotransferase (AST), alanine aminotransferase (ALT), gamma-glutamyl transferase (GGT) and urinary albumin-to-creatinine ratio (ACR).

**Diagnosis of CAN:** The evaluation included electrocardiograms for assessing the QTc interval, and cardiovascular reflex tests (CART) such as heart rate variability during deep breathing, the Valsalva maneuver, and responses to orthostatic changes. The results of CART were categorized as usual if no abnormal findings were detected. They were considered to indicate mild dysfunction if one out of the five tests was abnormal, moderate dysfunction if two or three of the tests were abnormal, and severe dysfunction if more than three tests were abnormal. All these measurements were performed using the ESP-01-PA Ewing Tester neuropathic measuring and analyzing system with an ECG module. At the same time, the diagnosis of CAN was evaluated by performing a sweat test using Sudoscan assessment. During the test, the patient places their hands and feet on the electrodes. The test takes 3 min to perform, is painless, and requires no subject preparation. Additionally, Sudoscan incorporates built-in algorithms that integrate electrochemical skin conductance with age to generate a score estimating the current risks of CAN (Sudoscan-CAN score) and chronic kidney disease (Sudoscan-Nephro score).

We have also calculated the SCORE2-Diabetes for the patients included in the study. SCORE2-Diabetes is a new algorithm developed, calibrated, and validated to predict 10-year risk of CVD in individuals with type 2 diabetes that enhances identification of individuals at higher risk of developing CVD across Europe [20]. Sex-specific competing risk-adjusted models were used, incorporating traditional risk factors, such as age, smoking, systolic blood pressure, total and HDL-cholesterol alongside diabetes-related variables including age at diabetes onset, glycated hemoglobin (HbA1c), and estimated glomerular filtration rate (eGFR) based on creatinine.

### 2.3. Statistical Analysis

The statistical analysis of the population was conducted using IBM SPSS v.20. Tests of normality used were Kolmogorov–Smirnov with a Lilliefors significance correction and Shapiro–Wilk statistic. Continuous variables usually distributed were presented as mean ± SD (standard deviation), and non-normal variables were expressed as median (interquartile range [IQR]). In contrast, categorical variables were reported as absolute counts and percentages. The *p*-value calculation for normally distributed variables was conducted using the ANOVA test, while for non-normally distributed variables, the Kruskal–Wallis test was applied. Statistical significance was determined at a 95% confidence interval. The χ^2^ test was utilized for categorical variables—multiple linear regression was used to estimate the independent correlation of the SCORE2-Diabetes risk with results of Sudoscan parameters.

## 3. Results

The cohort comprised 211 patients diagnosed with T2DM, without established ASCVD, 51.6% being male (n = 109), with a mean age of 58.35 ± 7.18 years and a mean weight of 90.36 ± 17 kg. The prevalence of CAN in our group was 67.2% (n = 142).

In the study group, according SCORE2-Diabetes, four patients (1.9%) were classified with moderate cardiovascular risk, thirty-five (16.6%) with high risk, and one hundred seventy-two (81.5%) with very high cardiovascular risk. Patients’ baseline characteristics are presented in Table 1.

Patients with very high cardiovascular risk were older and with longer diabetes duration. However, there were no notable differences in height, weight, or waist circumference across risk categories. Elevated levels of FPG and HbA1c were observed in higher-risk groups. Lipid levels, including TC, HDL-c, TGL, and LDL-c, did not significantly vary among risk groups. Lower eGFR was decreased parallel with increased CVR categories. While GGT showed a marginal difference across risk groups, overall, factors such as age, diabetes duration, FPG, HbA1c, and eGFR were identified as key indicators of cardiovascular risk in diabetic patients (Table 1).

There are significant variations in SBP while lying down among the three risk categories, with higher levels detected in higher-risk groups (*p* = 0.041). There are no notable differences in DBP in the lying position (*p* = 0.065). There are no substantial differences in SBP and DBP while standing across the risk categories (*p* > 0.05). During handgrip exercises, SBP does not significantly differ across risk groups (*p* = 0.494), yet there is a marginal difference in DBP (*p* = 0.073) and a significant difference in heart rate (*p* = 0.046) (Table 2).

The statistical analysis of Sudoscan’s parameters in relation to cardiovascular risk reveals significant differences. Specifically, the Sudoscan CAN-score exhibits significant variance among risk groups (*p* = 0.002), indicating a potential correlation between sudomotor dysfunction and cardiac autonomic neuropathy in individuals at higher risk of cardiovascular complications. Similarly, the Sudoscan Nephro-score displays a notable difference across risk categories, with lower scores observed in higher-risk groups (*p* = 0.001), suggesting a potential link between sudomotor dysfunction and renal function in individuals with increased cardiovascular risk. However, scores for Sudoscan parameters related to the feet and hands do not show significant differences across risk categories. These findings suggest that Sudoscan-derived measures may serve as valuable indicators of cardiovascular risk, particularly in assessing cardiac autonomic function and nephropathy in diabetic populations (Table 3).

The statistical analysis of the frequency of chronic diabetes complications based on cardiovascular risk indicates significant differences among the various risk categories. For cardiovascular autonomic neuropathy (CAN), the prevalence increases with the degree of cardiovascular risk, with the highest frequency observed in those with a risk greater than 20% (69.80%). The same trend is observed for diabetic polyneuropathy (DPN), chronic kidney disease (CKD), and diabetic retinopathy (DR), where the frequency of complications significantly increases with higher cardiovascular risk. These findings underscore the importance of proper monitoring and management of cardiovascular risk in addressing chronic diabetes complications (Table 4).

### Correlation of Sudoscan with SCORE 2—Diabetes

The scatterplot shows the relationship between the Sudoscan CAN-score, Sudoscan Nephro-score and the SCORE2-Diabetes. Sudoscan CAN-score showed a significantly positive correlation with SCORE 2-Diabetes and, Sudoscan Nephro-score score showed a significantly negative correlation with SCORE 2-Diabetes (Figure 1a,b).

The scatterplot shows the relationship between the Sudoscan feets-score, Sudoscan hands-score and the SCORE2-Diabetes. Sudoscan feets-score and Sudoscan hands-score showed a significantly negative correlation with SCORE 2-Diabetes (Figure 2a,b).

The area under the receiver operating characteristic (ROC) curve of the Sudoscan-CAN score to predict very high cardiovascular risk was 0.657 (95%CI: 0.569–0.745) of the total square (Figure 3). The Sudoscan-CAN score cut-off was 39.5, and the test had 34.3% sensitivity and 79% specificity to detect very high cardiovascular risk.

The multiple linear regression analysis examining clinical factors associated with SCORE2-Diabetes in patients with T2DM reveals several significant associations. Age and diabetes duration exhibit positive correlations with SCORE2-Diabetes, with standardized β-coefficients of 0.413 and 0.179, respectively, both statistically significant (*p* < 0.001). Similarly, HbA1c, LDL-c Sudoscan feet score, Sudoscan CAN-score, Ewing test score, and SBP in the supine position also demonstrate positive associations with SCORE2-Diabetes, with significant *p*-values (<0.05). Conversely, eGFR and Sudoscan Nephro-score display negative associations with SCORE2-Diabetes, suggesting that higher eGFR and Sudoscan Nephro-scores are associated with lower SCORE2-Diabetes values. These findings highlight the importance of age, diabetes duration, HbA1c, LDL-c, Sudoscan parameters, eGFR, and SBP in predicting SCORE2-Diabetes in patients with T2DM (Table 5).

## 4. Discussion

In this study, we analyzed the correlation between the Sudoscan-CAN score and cardiovascular risk, as calculated using the SCORE2-Diabetes algorithm. The aim was to investigate the potential association between sudomotor dysfunction and increased cardiovascular risk in individuals with Type 2 Diabetes Mellitus (T2DM).

The statistical assessment of Sudoscan parameters in the context of cardiovascular risk showed substantial differences. Notably, the Sudoscan CAN-score varied significantly across risk categories, suggesting an association between sudomotor dysfunction and cardiac autonomic neuropathy in patients at elevated risk for cardiovascular events. Multiple linear regression analysis revealed significant positive correlations between Sudoscan CAN-score and SCORE2-Diabetes in T2DM patients (*p* < 0.001).

The findings of the study highlight the significant cardiovascular risk faced by patients diagnosed with T2DM. The prevalence of CAN was significant within our studied patient cohort, being 67.2%. What is important to mention is that 69.8% of patients with very high cardiovascular risk also associate CAN. With a substantial portion of the cohort (81.5%) classified as having very high cardiovascular risk according to SCORE2-Diabetes, it underscores the critical need for comprehensive risk assessment and management strategies in this population.

In a meta-analysis conducted in 2003, Maser et al. synthesized the evidence base to evaluate the relationship between CAN and the risk of mortality in diabetes. CAN was associated with future risk of mortality both in cases of definite CAN and possible CAN, with a stronger association observed in definite CAN cases [21].

Also, Ewing et al. illustrated a 2.5-year mortality rate of 27.5%, which escalated by 25.5% after 5 years in patients with diabetes and definite CAN [22]. This stands in stark contrast to patients with diabetes and a normal autonomic function test (AFT), who exhibited a mortality rate of only 15% over the same 5-year period. Additionally, CAN also serves as a prognostic indicator for cardiovascular events (CVE) and mortality in the context of intensive glycemic control in type 2 diabetes. This was evidenced by the Action in Diabetes and Vascular Disease: Preterax and Diamicron Modified Release Controlled Evaluation (ADVANCE), Veterans Affairs Diabetes Trial (VADT), and ACCORD Studies [23,24,25].

Similar investigations, such as the study by T Yuan et al., have highlighted the utility of Sudoscan in formulating a cardiac risk score to evaluate cardiovascular autonomic neuropathy in asymptomatic diabetic patients [26]. Our findings align with these observations, bolstering the clinical relevance of Sudoscan in identifying significant cardiovascular risks among individuals with type 2 diabetes mellitus. While this study focused on creating a predictive model for asymptomatic patients, our research applies Sudoscan measurements directly to established cardiovascular risk scoring systems like SCORE2-Diabetes, providing a practical and efficient tool for assessing risk in a clinical setting. This integration of Sudoscan into routine assessments could facilitate earlier and more personalized interventions, enhancing preventive strategies and patient outcomes in diabetic cardiovascular care.

The research conducted by II Hussein et al. on the assessment of sudomotor function in hypertensive patients with or without T2DM underscores the importance of Sudoscan in evaluating cardiovascular risks [27]. Their findings support our study’s conclusions about the predictive relevance of Sudoscan scores for cardiovascular complications in diabetic patients. Both studies highlight the potential of Sudoscan to serve as a vital component of cardiovascular risk management, advocating for its integration into routine clinical practices to enhance early detection and preventive strategies in high-risk patient groups.

Our study introduces the utilization of the Ewing Test and Sudoscan Cardiovascular Autonomic Neuropathy Score as adjunctive tools for assessing cardiovascular risk in T2DM patients. This integration of novel assessment methods provides a more comprehensive evaluation beyond traditional risk scoring algorithms, potentially offering additional insights into cardiovascular health in this population.

Similar investigations, such as those by Vinik et al., have also highlighted the prognostic value of CAN assessments in predicting cardiovascular events in diabetic patients. Our findings parallel these studies, reinforcing the clinical utility of Sudoscan in detecting significant cardiovascular risk. Unlike conventional risk assessment tools, Sudoscan offers a quick and patient-friendly means to assess risk, potentially leading to earlier interventions [26].

The observed correlations between Sudoscan CAN-score and Sudoscan Nephro-score with SCORE2-Diabetes shed light on the interplay between autonomic neuropathy, renal function, and cardiovascular risk in T2DM patients. The positive correlation of Sudoscan CAN-score with cardiovascular risk suggests a potential association between autonomic dysfunction and increased cardiovascular risk. The identification of significant associations between various clinical parameters and SCORE2-Diabetes underscores the multifactorial nature of cardiovascular risk in T2DM patients. This highlights the importance of comprehensive risk stratification strategies that consider not only traditional risk factors but also novel markers such as autonomic function and renal health in guiding therapeutic interventions and optimizing outcomes.

The findings regarding the correlation between Sudoscan feets-score and Sudoscan hands-score with SCORE2-Diabetes suggest a potential role for Sudoscan as a non-invasive tool for assessing peripheral neuropathy in T2DM patients and predicting cardiovascular risk.

The clinical implications of these findings are significant, indicating that measuring sudomotor dysfunction using Sudoscan effectively identifies Type 2 Diabetes Mellitus (T2DM) patients who are at high cardiovascular risk. The inclusion of the Ewing Test and Sudoscan scores as part of cardiovascular risk assessments could serve as a valuable addition to traditional methods, potentially improving early detection and management strategies. This supports the integration of innovative diagnostic tools into routine clinical practice, which could enhance the proactive management of cardiovascular risks in diabetic populations, ultimately improving patient outcomes.

Further research is warranted to elucidate the clinical utility of Sudoscan scores in risk stratification and guiding therapeutic interventions in this population.

### Limit

While the study provides valuable insights into the relationship between Ewing Test, Sudoscan scores, and cardiovascular risk in T2DM patients, certain limitations need to be acknowledged. Firstly, these include the relatively small sample size and cross-sectional design. Secondly, all participants included in our research are from Romania, a very-high-risk European region in terms of CV mortality [27]. Also, the study enrolled patients from a single center.

Future prospective studies with larger sample sizes and longitudinal follow-up are needed to validate these findings and explore the utility of novel assessment tools in improving cardiovascular outcomes in T2DM patients.

## 5. Conclusions

Our study confirms the significant role of sudomotor dysfunction, as measured by Sudoscan, in identifying T2DM patients at high cardiovascular risk. The use of the Ewing Test and Sudoscan scores offers a promising adjunctive tool in the cardiovascular risk assessment of this population. These findings underscore the need for incorporating novel diagnostic tools into standard practice to enhance the early detection and management of cardiovascular risk among diabetic patients.

## Figures and Tables

**Figure 1 medicina-60-00828-f001:**
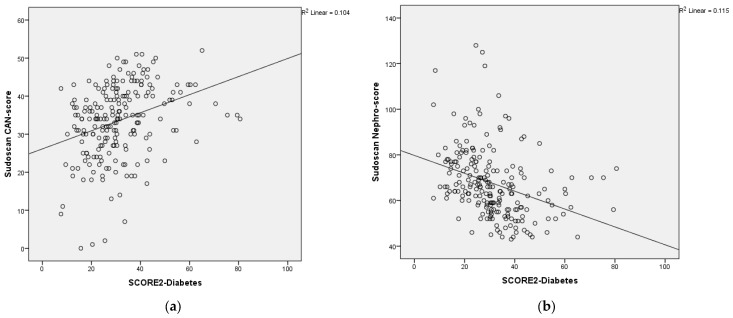
(**a**) Scatterplot showing the relationship between Sudoscan CAN-score on *y*-axis and SCORE 2—Diabetes on *x*-axis. (**b**) Scatterplot showing the relationship between Sudoscan Nephro-score on *y*-axis and SCORE 2—Diabetes on *x*-axis.

**Figure 2 medicina-60-00828-f002:**
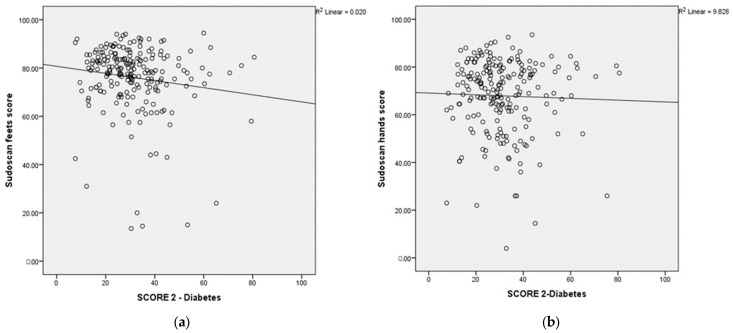
(**a**) Scatterplot showing the relationship between Sudoscan feets-score on *y*-axis and SCORE 2—Diabetes on *x*-axis. (**b**) Scatterplot showing the relationship between Sudoscan hands-score on *y*-axis and SCORE 2—Diabetes on *x*-axis.

**Figure 3 medicina-60-00828-f003:**
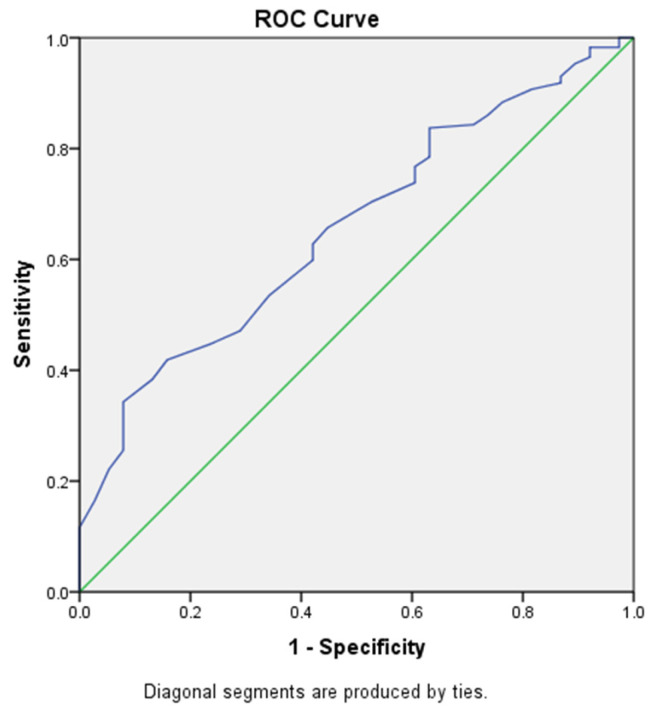
ROC curve of Sudoscan-CAN score in detecting very high cardiovascular risk in patients with T2DM.

**Table 1 medicina-60-00828-t001:** Anthropometric and biochemical parameters in relation to cardiovascular risk.

Cardiovascular Risk	(5–10%) n = 4		(10–20%) n = 35		(>20%) n = 172		Total		
	Mean	Std. Deviation	Mean	Std. Deviation	Mean	Std. Deviation	Mean	Std. Deviation	*p*-Value
**Age (years)**	**45**	**4.92**	**52**	**5.27**	**60**	**6.59**	**58**	**7.18**	**<0.001**
**Diabetes duration (years) ***	**6.50**	**8**	**5**	**5**	**7.5**	**8**	**7**	**8**	**0.036**
Height (cm)	170.00	8.98	167.14	11.00	166.47	8.71	166.64	9.10	0.701
Weight (kg)	95.00	20.70	88.86	17.42	90.55	16.91	90.36	17.00	0.745
WC (cm)	110.25	21.82	105.49	11.98	106.92	12.94	106.74	12.92	0.722
**FPG (mg/dL)**	**200.50**	**43.93**	**162.11**	**61.97**	**206.06**	**91.39**	**198.67**	**87.85**	**0.025**
**HbA1c (%)**	**7.26**	**0.52**	**7.17**	**1.19**	**8.61**	**2.04**	**8.34**	**1.99**	**<0.001**
TC (mg/dL)	212.25	44.63	184.89	50.93	200.75	57.15	198.34	56.09	0.277
HDL-c (mg/dL)	54.10	12.55	47.12	14.40	50.32	13.08	49.86	13.30	0.352
LDL-c (mg/dL)	94.63	9.31	101.85	45.22	109.13	50.53	107.73	49.32	0.668
TGL (mg/dL) *	292.5	612.5	173	169	179	132.5	178	138	0.103
**eGFR (mL/min/1.73 m^2^)**	**98.43**	**17.87**	**92.74**	**30.69**	**74.61**	**24.27**	**78.07**	**26.27**	**<0.001**
GGT (UI/L) *	52.50	62	34	73	38	50	37.5	52	0.05
ACR (mg/g) *	23.13	39.37	12	28.54	31.25	71.33	26.27	49.12	0.78
B12 vitamine (pg/mL) *	889.5	91	427.5	158	350	269	382	364	0.057

Abbreviations: WC = waist circumference, FPG = fasting plasma glucose, HbA1c = glycated hemoglobin, TC = total cholesterol, HDL-c = high-density lipoprotein, LDLc = low-density lipoprotein, TGL = triglycerides, eGFR = estimated glomerular filtration rate, GGT = gamma-glutamyl transferase, ACR = Albumin-to-creatinine ratio, * variables expressed as median, interquartile range [IQR], statistical significance, *p* < 0.05.

**Table 2 medicina-60-00828-t002:** Ewing’s parameters in relation to cardiovascular risk.

Cardiovascular Risk	(5–10%) n = 4		(10–20%) n = 35		(>20%) n = 172		Total		
	Mean	Std. Deviation	Mean	Std. Deviation	Mean	Std. Deviation	Mean	Std. Deviation	*p*-Value
**SBP supine position (mmHg)**	**122.75**	**21.22**	**127.29**	**17.54**	**134.83**	**18.13**	**133.35**	**18.27**	**0.041**
DBP supine position (mmHg)	88.00	21.86	74.57	9.41	77.10	11.19	76.89	11.24	0.065
HR supine position (bpm)	82.75	10.01	73.23	9.55	76.17	11.00	75.81	10.81	0.147
SBP standing (mmHg)	125.50	10.79	123.69	12.96	129.14	19.91	128.17	18.86	0.286
DBP standing (mmHg)	86.50	15.80	74.86	9.50	76.49	12.31	76.41	12.00	0.181
HR standing (bpm)	84.25	9.22	76.34	10.81	80.00	11.94	79.47	11.77	0.176
SBP Handgrip (mmHg)	137.75	18.12	143.40	16.04	145.72	16.48	145.18	16.41	0.494
DBP Handgrip (mmHg)	89.00	15.87	85.20	14.04	81.10	10.74	81.93	11.53	0.073
**HR Handgrip (bpm)**	**90.75**	**10.28**	**80.69**	**9.85**	**85.05**	**10.89**	**84.44**	**10.82**	**0.046**
HRVi index	26.00	6.78	19.41	7.01	16.59	7.41	17.25	7.48	0.653
Ewing score *	2	5	3	2	4	3	3	3	0.194

Abbreviations: SBP = systolic blood pressure, DBP = diastolic blood pressure, HR = heart rate, HRVi = heart rate variability index, * variables expressed as median, interquartile range [IQR], statistical significance, *p* < 0.05.

**Table 3 medicina-60-00828-t003:** Sudoscan’s parameters in relation to cardiovascular risk.

Cardiovascular Risk	(5–10%) n = 4		(10–20%) n = 35		(>20%) n = 172		Total		
	Mean	Std. Deviation	Mean	Std. Deviation	Mean	Std. Deviation	Mean	Std. Deviation	*p*-Value
**Sudoscan Nephro-score**	**90.00**	**24.59**	**72.34**	**9.29**	**66.06**	**15.39**	**67.56**	**15.19**	**0.001**
**Sudoscan CAN-score ***	**16.5**	**28**	**31**	**12**	**35**	**13**	**34.5**	**13**	**0.002**
Sudoscan left feet score (uS)	76.00	21.46	77.46	10.78	75.28	14.89	75.65	14.37	0.717
Sudoscan right feet score (uS)	73.50	24.53	76.77	11.28	76.66	13.78	76.62	13.57	0.898
Sudoscan left hand score (uS)	57.75	22.90	71.63	13.19	68.66	15.87	68.95	15.63	0.209
Sudoscan right hand score (uS)	50.75	20.09	69.80	13.23	66.72	16.12	66.93	15.88	0.069

* variables expressed as median, interquartile range [IQR], Statistical significance, *p* < 0.05.

**Table 4 medicina-60-00828-t004:** The frequency of chronic diabetes complications based on cardiovascular risk.

Cardiovascular Risk	5–10% (n = 4)	%	10–20% (n = 35)	%	>20% (n = 172)	%
**CAN**	2	50.00	20	57.10	120	69.80
**DPN**	2	50.00	19	54.29	102	59.30
**CKD**	1	25.00	9	25.71	85	49.42
**DR**	2	50.00	7	20.00	64	37.21
**PAD**	0	0.00	0	0.00	34	19.77

Abbreviations: CAN = cardiovascular autonomic neuropathy, CKD = chronic kidney disease, DR = diabetic retinopathy, DPN = diabetic polyneuropathy, PAD = peripheral artery disease.

**Table 5 medicina-60-00828-t005:** Clinical factors associated with SCORE2-Diabetes in patients with T2DM using multiple linear regression.

	Standard β-Coefficient	95% CI	*p*-Value
**Age**	0.804	[0.585, 1.022]	**<0.001**
**Diabetes duration**	0.627	[0.345, 0.908]	**<0.001**
**HbA1c**	2.437	[1.617, 3.258]	**<0.001**
**LDL-c**	0.048	[0.014, 0.083]	**<0.001**
**eGFR**	−0.225	[−0.285, −0.165]	**<0.001**
**Sudoscan Nephro-score**	−0.279	[−0.389, −0.169]	**<0.001**
**Sudoscan CAN-score**	0.405	[0.233, 0.576]	**<0.001**
**Ewing test score**	1.102	[0.327, 1.878]	**0.006**
**SBP supine position**	0.169	[0.075, 0.262]	**<0.001**

Abbreviations: HbA1c = glycated hemoglobin, LDLc = low-density lipoprotein, eGFR = estimated glomerular filtration rate, SBP = systolic blood pressure, statistical significance, *p* < 0.05.

## Data Availability

The data presented in this study are available on request from the corresponding author. The data are not publicly available due to the hospital’s privacy policy.

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
