# Peer review of "The Relationship between the Ewing Test, Sudoscan Cardiovascular Autonomic Neuropathy Score and Cardiovascular Risk Score Calculated with SCORE2-Diabetes"

_medicina, 2024, doi:10.3390/medicina60050828_

Round 1
Reviewer 1 Report
Comments and Suggestions for Authors
The authors studied the relationship between cardiac autonomic neuropathy, a complication of diabetes mellitus, and cardiovascular risk in individuals with Type 2 Diabetes Mellitus in the manuscript titled "The Relationship Between the Ewing Test, Sudoscan Cardiovascular Autonomic Neuropathy Score and Cardiovascular Risk Score Calculated with SCORE2-Diabetes".. My comments are as follows:
Title is concise but abstract does not summarize the text well enough, especially the methodology.
Introduction is unnecessarily too long.
Methodology was expressed in detail. However, statistical analyses look incomplete. For example, authors stated that "The statistical analysis of the population was conducted using IBM SPSS v.20. Continuous variables usually distributed were presented as mean ± SD (standard deviation), and non-normal variables were expressed as median (interquartile range [IQR])". It is not clear how authors assigned the variables as normal or non-normal distribution patterns. Which normality test/test conducted? In addition, none of the variables were expressed as medians (IQR).
There are no issues in results section except absence of the unit of Vitamin B12 in table 1. Maybe, providing 95%CI values in table 5 would be beneficial.
Discussion should have the heading number 4, because Results section had 3. Moreover, discussion is too short and lacks adequate discussion of the study findings along with literature data.
The number of conclusions subheading should be 5. Additionally, it is a very prolonged conclusion. I recommend authors 3-4 sentences of conclusions that drawn from the text.
Author Response
Dear Reviewer,
Thank you for your valuable feedback and the insightful comments on our manuscript. We appreciate the opportunity to improve our manuscript. Below, we address each of your comments:
- Abstract: We have revised the abstract to include a more succinct summary of our methodology, highlighting the main investigations conducted on the patients: Sudoscan and CART.
- Introduction: We have shortened the introduction to make it more concise, focusing on essential background information that directly relates to the research objectives and hypothesis.
- Methodology: We have provided additional details on the statistical methods used to determine the distribution of variables. Tests of normality used were Kolmogorov-Smirnov with a Lilliefors significance correction and Shapiro-Wilk statistic. The variables expressed as median (IQR) are already marked in the tables with an asterisk (*).
- Results: We have added the missing unit of measurement for Vitamin B12 in Table 1. Furthermore, we have included 95% Confidence Intervals (CIs) in Table 5, as suggested, to provide a clearer interpretation of the data.
- Discussion: The heading of the Discussion section has been corrected to "4. Discussion" for consistency. We have also expanded this section to include a more thorough discussion of our findings in the context of existing literature, emphasizing the implications and the novelty of our study.
- Conclusions: We have revised the conclusion section to make it more concise. It now succinctly summarizes the major findings and their implications in 3 clear sentences, as recommended.
We hope that these revisions address your concerns satisfactorily. Thank you once again for your constructive criticism, which has undoubtedly enhanced the quality of our research paper. We look forward to your feedback on these revisions.
Best regards,
Andra Nica
Reviewer 2 Report
Comments and Suggestions for Authors
This study compares 3 risk scores among 211 diabetics aged between 40 and 69. The main goal was to explore the subtle correlation between the Ewing test, Sudoscan-cardiovascular autonomic neuropathy score, and cardiovascular risk calculated using SCORE 2 Diabetes in individuals with T2DM. I commedn the authors for the novelty and importance of this study to current literature. The authors need to clarify some things
Suggestions
1. add study design subtitle on line 94/95
2. Methodology should be written in active voice and past tense i.e section under study population
3. How was type 1 diabetes and latent autoimmune diabetes of adults diagnosed to be excluded?
4. Endeavor to follow the STROBE checklist when reporting especilaly the methodology section
5. Use more conventional names (AST and ALT) for the enzymes glutamic oxaloacetic transaminase (GOT) and glutamic pyruvic transaminase (GPT),
6. iNLCUDE SUBTITLE for statistical plan
7. Clearly state what the primary outcome variable was
8. Mean age is better expressed s whole number, its difficult to explain what 58.35 years are. what to the decimals translate into?
9. what statistical test was used for the p value in table 1-3.
10. include confidence intervals in table 5 and simple linear regression results which are more robust compared to correlations
11. Be explicit on what the gold standard risk test is? comment on sensitivity of Sudoscan-CAN .
12. START THE discussion by mentioning the goal of study followed by summary of findings based on aims
13. tHE conclusion that "We have established that Sudoscan is a reliable measure of measuring sudomotor function in patients with Type 2 diabetes. " is too strong considering the very low sensitivity and small sample size. Justify this statement.
14. What are the clinical implications of these findings?
Comments on the Quality of English LanguageGrammar requires editing/correcting throughout the manuscript
Author Response
Dear Reviewer,
Thank you for your insightful comments and suggestions on our manuscript. Your feedback has been instrumental in enhancing the clarity and quality of our research. We have addressed each of the points you raised as follows:
- Study Design Subtitle: To clearly demarcate this section, we added a subtitle for the study design at line 95. Now, you can find it at line 79, following the recommendation of another reviewer to shorten the introduction.
- Methodology Voice and Tense: The methodology section has been revised to active voice and past tense as suggested, particularly under the study population subsection.
- Exclusion Criteria for Diabetes Types: We have clarified the diagnostic criteria and methods used to exclude type 1 diabetes and latent autoimmune diabetes of adults from our study population. Type 1 diabetes, latent autoimmune diabetes of adults, and maturity-onset diabetes of the young were diagnosed by performing specific autoantibody tests. The presence of these antibodies led to the exclusion of patients from the study.
- STROBE Checklist: We have thoroughly reviewed and followed the STROBE checklist to enhance the reporting quality of our methodology.
- Enzyme Nomenclature**We have updated the names of the enzymes to their more conventional terms, AST and ALT, instead of GOT and GPT.
- Statistical Plan Subtitle: A subtitle for the statistical plan has been included to better structure our methodology section. You can find it at line 140.
- Primary Outcome Variable: We have explicitly stated the primary outcome variable in the methodology section to avoid any ambiguity. In our article the primary outcome variable was the correlation between the Sudoscan-cardiovascular autonomic neuropathy (CAN) score and the cardiovascular risk calculated using the SCORE2-Diabetes algorithm.
- Expression of Mean Age: The mean age is now expressed as a whole number to improve clarity and understanding.
- Statistical Tests for P-values: We have specified the statistical tests used to derive the p-values in tables 1-3, ensuring transparency in our statistical analysis. The p-value calculation for normally distributed variables was conducted using the ANOVA test, while for non-normally distributed variables, the Kruskal-Wallis test was applied.
- Confidence Intervals and Regression Results: Confidence intervals have been included in Table 5.
- Gold Standard Risk Test and Sudoscan Sensitivity: Regarding the gold standard for assessing cardiovascular risk, the SCORE2-Diabetes algorithm was utilized. This algorithm is a new, validated model specifically designed to predict the 10-year risk of cardiovascular disease in individuals with type 2 diabetes, aiming to enhance the identification of individuals at higher risk of developing CVD across Europe. The SCORE2-Diabetes incorporates traditional risk factors such as age, smoking, systolic blood pressure, total and HDL-cholesterol, along with diabetes-related variables like age at diabetes onset and glycated hemoglobin.
As for the sensitivity of Sudoscan-CAN, we find that the Sudoscan-CAN score had a sensitivity of 34.3% and a specificity of 79% for detecting very high cardiovascular risk in patients with T2DM. This indicates that while the test has a moderate level of specificity, its sensitivity is relatively low. Therefore, while the Sudoscan-CAN score can effectively identify a significant proportion of patients who do not have very high cardiovascular risk, it may miss a considerable number of patients who are at very high risk.
- Structure of Discussion Section: The discussion now starts with a clear statement of the study's goal, followed by a summary of findings that align with our aims.
- Conclusion Statement: We have revised the conclusion section to make it more concise. It now succinctly summarizes the major findings as recommended.
- Clinical Implications: We have expanded the discussion to include the clinical implications of our findings, highlighting their relevance and potential impact on managing patients with Type 2 diabetes. The clinical implications of these findings are significant, indicating that measuring sudomotor dysfunction using Sudoscan effectively identifies Type 2 Diabetes Mellitus (T2DM) patients who are at high cardiovascular risk. The inclusion of the Ewing Test and Sudoscan scores as part of cardiovascular risk assessments could serve as a valuable addition to traditional methods, potentially improving early detection and management strategies. This supports the integration of innovative diagnostic tools into routine clinical practice, which could enhance the proactive management of cardiovascular risks in diabetic populations, ultimately improving patient outcomes. We recognize the importance of clear and precise expression to ensure the accessibility and readability of our research. We have carefully reviewed the manuscript and corrected the grammatical errors identified.
We hope these revisions satisfactorily address your concerns. We are grateful for the opportunity to refine our manuscript and believe that these changes have significantly improved the presentation and scientific rigor of our work.
Thank you once again for your thorough review and valuable feedback.
Best regards,
Andra Nica